# 5-FU-miR-15a Inhibits Activation of Pancreatic Stellate Cells by Reducing YAP1 and BCL-2 Levels In Vitro

**DOI:** 10.3390/ijms24043954

**Published:** 2023-02-16

**Authors:** Vanessa M. Diaz Almanzar, Kunal Shah, Joseph F. LaComb, Aisharja Mojumdar, Hetvi R. Patel, Jacky Cheung, Meiyi Tang, Jingfang Ju, Agnieszka B. Bialkowska

**Affiliations:** 1Department of Medicine, Renaissance School of Medicine at Stony Brook University, Stony Brook, NY 11794, USA; 2Department of Pathology, Renaissance School of Medicine at Stony Brook University, Stony Brook, NY 11794, USA

**Keywords:** pancreatic stellate cells, chronic pancreatitis, microRNA

## Abstract

Chronic pancreatitis is characterized by chronic inflammation and fibrosis, processes heightened by activated pancreatic stellate cells (PSCs). Recent publications have demonstrated that miR-15a, which targets *YAP1* and *BCL-2*, is significantly downregulated in patients with chronic pancreatitis compared to healthy controls. We have utilized a miRNA modification strategy to enhance the therapeutic efficacy of miR-15a by replacing uracil with 5-fluorouracil (5-FU). We demonstrated increased levels of YAP1 and BCL-2 (both targets of miR-15a) in pancreatic tissues obtained from *Ptf1aCre^ERTM^* and *Ptf1aCre^ERTM^;LSL-Kras^G12D^* mice after chronic pancreatitis induction as compared to controls. In vitro studies showed that delivery of 5-FU-miR-15a significantly decreased viability, proliferation, and migration of PSCs over six days compared to 5-FU, TGFβ1, control miR, and miR-15a. In addition, treatment of PSCs with 5-FU-miR-15a in the context of TGFβ1 treatment exerted a more substantial effect than TGFβ1 alone or when combined with other miRs. Conditioned medium obtained from PSC cells treated with 5-FU-miR-15a significantly inhibits the invasion of pancreatic cancer cells compared to controls. Importantly, we demonstrated that treatment with 5-FU-miR-15a reduced the levels of YAP1 and BCL-2 observed in PSCs. Our results strongly suggest that ectopic delivery of miR mimetics is a promising therapeutic approach for pancreatic fibrosis and that 5-FU-miR-15a shows specific promise.

## 1. Introduction

Chronic pancreatitis is associated with a 50% mortality rate and is one of the most substantial risk factors for pancreatic cancer development [1]. Chronic inflammation and fibrosis leads to structural damage, which impairs the organ’s critical endocrine and exocrine functions [2,3]. Indeed, widespread fibrosis is the prominent histological feature of chronic pancreatitis and results from the progressive activation of pancreatic stellate cells (PSCs). The most common consequences of chronic pancreatitis includes recurrent or constant abdominal pain, diabetes mellitus (endocrine insufficiency), and maldigestion (exocrine insufficiency) [4]. Fundamentally, diagnosis of early-stage chronic pancreatitis is challenging as its features are shared with other disorders. Recent studies have shown that destroying pancreatic acinar cells induces a proinflammatory response that triggers PSCs activation, leading to widespread fibrosis [5].

Repeated pancreatic injury results in the dysregulation of pancreatic acinar cell function, which is typified by an increase in intracellular calcium, dysfunction of the endoplasmic reticulum and mitochondria, chronic activation of pancreatic enzymes, and alteration of lipid metabolic pathways [6,7,8]. These changes sequentially lead to impaired acinar cell autophagy. In addition, there is increased cell death, followed by immune cell infiltration and subsequent activation. This causes the activation of PSCs, resulting in the widespread fibrosis that characterizes chronic pancreatitis [6]. Importantly, PSCs play a crucial role in the progression of pancreatic cancer and support pancreatic cancer tumorigenicity and invasion [9]. During pancreatic insult, stimuli such as alcohol, autophagy, and inflammatory mediators trigger the activation of signaling pathways within quiescent PSCs. The crosstalk between these pathways, which includes MAPKs (ERK, JNK, and p38), PI3K, STAT3, NFκB, TGF-β, ROCK, AP-1, HIPPO, WNT, autophagy, and calcium signaling, results in quiescent PSCs acquiring a fibroblast phenotype [10,11,12,13,14,15,16,17,18,19,20]. These pathways mediate extensive extracellular matrix remodeling, collagen fibers deposition, and increased tissue stiffness. In addition, paracrine signals and autocrine pathways stimulate PSCs constantly [21].

Recent studies have demonstrated aberrant micro-RNA (miR) expression patterns in pancreatic tissues obtained from patients experiencing chronic pancreatitis or pancreatic cancer compared to tissues from unaffected individuals [22,23,24]. Bloomston et al. identified 22 miRs overexpressed and two suppressed in chronic pancreatitis compared to normal pancreatic tissue [23]. Bioinformatics analysis of miR networks and pancreatitis-associated genes provided potential therapeutic targets, e.g., hsa-miR-15a [25]. It has been demonstrated that miR-29 negatively regulates the PI3K-AKT pathway and suppresses fibrosis, while miR-200 blocks TGF-β signaling through ZEB1 and ZEB2 and inhibits epithelial-to-mesenchymal transition [26]. In contrast, miR-21 and miR-145 positively regulate TGF-β signaling and promote fibrosis. In addition, it has been shown that miR-15b and miR-16 induce apoptosis of active PSCs via the BCL-2 pathway [27]. Such miRs regulate cell proliferation, tissue remodeling, and migration [28]. Notably, some of the tested miRs, including miR-15a, target the HIPPO/YAP/TAZ signaling and autophagy pathways that regulate PSCs activation following pancreatic insult and, thus, likely modulate the fibrotic response [27]. Studies from Dr. Ju showed that ectopic delivery of miR-15a inhibits the growth of colorectal and pancreatic cells in in vitro and in vivo models and that a modified miR-15a, with 5-FU replacing uracil, has increased potency and feasibility of delivery by increasing the lipophilicity of the miR-15a [29,30,31]. Here, we demonstrate that 5-FU-miR-15a significantly reduces the growth and migration of PSCs by inhibiting inflammatory factors and effectively inhibits the invasion of pancreatic cancer cells in vitro.

## 2. Results

### 2.1. YAP1 and BCL-2 Are Upregulated, While miR-15a Is Downregulated during Chronic Pancreatitis

Recent publications demonstrate that miR-15a levels are downregulated in human chronic pancreatitis specimens compared to healthy controls [25]. To investigate the role of miR-15a in chronic pancreatitis, we first assessed miR-15a levels and its targets (YAP1 and BCL-2) upon the development of chronic pancreatitis in *Ptf1aCre^ERTM^* and *Ptf1aCre^ERTM^;LSL-KRAS^G12D^* mice. These two animal models allow us to analyze different stages of pancreatitis progression. We implemented a four-week cerulein regimen that leads to the development of mild chronic pancreatitis [32]. Pancreatic histology in *Ptf1a-Cre^ERTM^* control mice were similar to those found in human chronic pancreatitis patients. This is characterized by inflammatory infiltrates, acinar cell atrophy, edema, and fibrosis. Notably, in *Ptf1aCre^ERTM^;LSL-Kras^G12D^* mice, the same injury regimen leads to early neoplasia, characterized by extensive fibrosis and pancreatic exocrine insufficiency [33,34,35].

Immunohistochemical analysis of YAP1 and BCL-2 showed that both factors were upregulated in pancreatic tissues of *Ptf1aCre^ERTM^* and *Ptf1aCre^ERTM^;LSL-KRAS^G12D^* mice treated with cerulein as compared to appropriate controls (Figure 1A and Figure 1B, respectively). Furthermore, the increase of both YAP1 and BCL-2 was observed in injured pancreatic acinar cells, early pancreatic neoplastic cells, and fibrotic microenvironment upon cerulein treatment (Figure 1A,B). The overall increase in staining of YAP1 and BCL-2 was connected to the increase in the fibrotic component and injured pancreatic acinar cells that undergo acinar-to-ductal metaplasia toward early pancreatic neoplasia.

Furthermore, we found that miR-15a expression was significantly decreased in the pancreas of *Ptf1aCre^ERTM^* and *Ptf1aCre^ERTM^;LSL-KRAS^G12D^* mice treated with cerulein as compared to the control (PBS-treated mice) (Figure 2A). In contrast, the levels of mRNA of *Yap1* and *Bcl2* were markedly increased in *Ptf1aCre^ERTM^* and *Ptf1aCre^ERTM^;LSL-KRAS^G12D^* mice treated with cerulein as compared to the appropriate controls (Figure 2B,C). Interestingly, compared with PBS-treated controls, the increase in *Yap1* in the pancreas revealed a 6-fold and 14-fold upregulation in *Ptf1aCre^ERTM^* change in *Ptf1aCre^ERTM^;LSL-KRAS^G12D^* mice under chronic pancreatitis conditions, respectively. Furthermore, we observed an increase in *Bcl2* levels with a 3-fold and 4-fold upregulation in *Ptf1aCre^ERTM^* and *Ptf1aCre^ERTM^;LSL-KRAS^G12^* mice, respectively, as compared to appropriate controls, respectively. In addition, we assessed the activation of PSCs by determining the levels of *Col1a1*, *Mmp9*, and *Il6* factors previously shown to be upregulated in PSCs during chronic pancreatitis [12,36,37,38,39,40,41]. Our results demonstrate that the markers mentioned above are upregulated in *Ptf1aCre^ERTM^* and *Ptf1aCre^ERTM^;LSL-KRAS^G12^* treated with cerulein as compared to the controls (PBS-treated mice) (Figure 2D–F). In addition, we assessed the protein levels of YAP1 and BCL-2 in *Ptf1aCre^ERTM^;LSL-KRAS^G12D^* after four weeks of PBS or CER treatment. Our results showed that there is a significant increase in both proteins upon chronic injury (Figure 2G).

Furthermore, we performed immunofluorescence staining of *Ptf1aCre^ERTM^;LSL-Kras^G12D^* mice treated with tamoxifen for one week and subsequently with PBS or CER for four weeks with antibody against αSMA and Ki-67, and included representative images in Figure 3A. As shown, there are αSMA^+^ cells that are positive for Ki-67, confirming the proliferative status of PSCs. In addition, we showed that there is a significant increase in fibrosis by Masson’s Trichrome staining in CER-treated *Ptf1aCre^ERTM^;LSL-Kras^G12D^* mice as compared to PBS-treated mice for four weeks (Figure 3B).

### 2.2. 5-FU-miR-15a Significantly Reduced the Proliferation of Murine Pancreatic Stellate Cells In Vitro

Recent publications demonstrated that exogenous delivery of miR-15a and, specifically, 5-FU-miR-15a significantly reduced the expression levels of *YAP1* and *BCL2* in pancreatic and colorectal cancer cell lines, respectively, and led to growth inhibition [29,31]. Therefore, we performed a Cell-Titer Glo assay to assess the efficacy of 5-FU-miR-15a on the viability of murine pancreatic stellate cells. For six days, PSCs were treated with variable concentrations of TGFβ1, 5-FU, control (CTRL) miR, miR-15a, and 5-FU-miR-15a (Figure 4). Our results showed that 5 ng/mL and 10 ng/mL TGFβ1 and 5-FU at 1 mM and 50 μM reduced the viability of PSCs. Importantly, 5-FU-miR-15a achieved the most potent inhibition of PSCs viability at nanomolar concentrations, while CTRL miR and miR-15a exert only a weak inhibitory effect.

TGFβ1 is a proinflammatory factor that stimulates the activation of PSCs during the development and progression of chronic pancreatitis towards early pancreatic neoplasia [42,43,44]. Its role in the increase in α-SMA and the production of extracellular matrix has been well established in PSCs [19]. In addition, TGFβ1 induces multiple pathways and growth factors, including FGF and PDGF, that might result in an increase in proliferation. However, multiple studies show that TGFβ1 may exert no effect on proliferation or show inhibitory effects [45]. Thus, to assess the effect of 5-FU-miR-15a on PSCs proliferation in more detail, we performed a time course analysis by treating cells as a single agent and in the context of TGFβ1. Our results showed that TGFβ1, 5-FU, and 5-FU-miR-15a reduced the viability of PSCs at each tested time point, with the last treatment demonstrating the most significant reduction in cell viability (Figure 5A–D) as compared to the controls. Simultaneously, treatment with miR-15a only slightly reduced the proliferation of PSCs. Notably, 5-FU-miR-15a in the context of TGFβ1 treatment manifested the most potent inhibitory effect while other treatments only slightly reduced PSCs proliferation (Figure 5E–H). Together, these results demonstrate that 5-FU-miR-15a alone and in the context of TGFβ1 treatment showed the most significant reduction in PSCs proliferation.

### 2.3. 5-FU-miR-15a Significantly Reduced Migration of Murine Pancreatic Stellate Cells and Invasion of Pancreatic Cancer Cells In Vitro

In addition to their capability to proliferate during the development and progression of chronic pancreatitis, PSCs can migrate [46,47,48]. To investigate the impact of 5-FU-miR-15a on the migration capacity of PSCs, we performed a “scratch” test. The results presented in Figure 6 and Figure 7 show that 5-FU-miR-15a reduced the migration of PSCs most significantly as compared to controls when used as a single agent and in combination with TGFβ1 12 h and 22 h after the scratch. It has also been shown that PSCs drive the development and progression of pancreatic cancer. Thus, we performed an invasion assay to investigate whether treatment of PSCs with 5-FU-miR-15a reduced their capability to stimulate pancreatic cancer cell invasion. Conditioned media obtained from PSCs treated with TGFβ1, 5-FU, CTRL miR, miR-15a, and 5-FU-miR-15a alone and in combination with TGFβ1 were used as chemoattractants for pancreatic cancer cell invasion. As shown in Figure 8A,B, 5-FU-miR-15a alone and in combination with TGFβ1 most efficiently reduced the invasion of pancreatic cancer cells as compared to appropriate controls.

### 2.4. 5-FU-miR-15a Significantly Reduced Inflammatory Markers in Pancreatic Stellate Cells

During the development of chronic pancreatitis, multiple signaling pathways are activated in PSCs [16,19,49]. In addition, miR-15a has been shown to impact the levels of inflammatory markers in various diseases [50,51,52,53]. Thus, we investigated the ability of 5-FU-miR-15a to alter expression levels of the components of these pathways. We treated PSCs with DMSO (vehicle) and 5 ng/mL TGFβ1, tested compounds (5-FU, CTRL miR, miR-15a, and 5-FU-miR-15a) at 50 nM concentration, and collected the cells for RNA and protein analysis after 24 h. We showed a slight increase in the expression levels of *Yap1*, *Bcl-2*, and *Mmp9* and high induction of *Il6* levels upon TGFβ1 stimulation (Figure 9A–D). By itself, 5-FU at the tested concentration only marginally inhibited *Yap1*, *Mmp9*, and *Il6*. We have noticed that miR-15a and 5-FU-miR-15a can reduce mRNA levels of *Yap1*, *Bcl-2*, *Mmp9*, and *Il6*, with 5-FU-miR-15a having the most significant impact. Western blot analysis confirmed the inhibitory effect of 5-FU-miR-15a on YAP1 and BCL-2 by itself (Figure 9A) and during co-treatment with TGFβ1 (Figure 9B).

### 2.5. 5-FU-miR-15a Significantly Reduced Proliferation of Human Pancreatic Stellate Cells and Expression of YAP1 and BCL-2 In Vitro

We tested three commercially available PSCs of human origin (please see Materials and Methods Section) to investigate whether 5-FU-miR-15a has similar efficacy in inhibiting the proliferation of murine and human PSCs. Our results demonstrate that treatment with 5-FU-miR-15a most effectively reduces the proliferation of three human pancreatic stellate cells compared to appropriate controls (Figure 10). Importantly, 5-FU-miR-15a exerts its inhibitory effects alone and with TGFβ1 co-treatment in all tested cell lines.

In addition, to determine the effect of 5-FU-miR-15a on the expression of YAP1 and BCL-2, we treated three hPSCs with DMSO (vehicle) and 5 ng/mL TGFβ1, tested compounds (5-FU, CTRL miR, miR-15a, and 5-FU-miR-15a) at 50 nM, and collected the cells for RNA and protein analysis after 24 h. RNA and protein analyses showed that 5-FU-miR-15a most significantly decreases the levels of YAP1 and BCL-2 as compared to the controls (Figure 11).

## 3. Discussion

Chronic pancreatitis is a fibroinflammatory disease that results in fibrosis and atrophy of the pancreas, and dysfunction of endocrine and exocrine activity of the organ [1,3,4,8,54]. It has been demonstrated that PSCs play a vital role in the development of fibrosis and its progression and, thus, impacts pancreatic acinar cells and immune cell activation [10,12,19,49,55,56,57,58]. Chronic pancreatitis is a risk factor for pancreatic cancer, and mutations in the proto-oncogene *KRAS* are found in nearly all cases of pancreatic cancer [59,60,61]. Importantly, mutations of *KRAS* are often found in early neoplastic lesions resulting from chronic pancreatitis. Therefore, it has been suggested that elevated Kras levels in combination with a persistent inflammatory injury during chronic pancreatitis lead to early neoplasia and increase the likelihood of developing pancreatic cancer [62,63]. Currently, none of the limited treatments address the development and progression of fibrosis due to the activation of PSCs. Due to the limited availability of chronic pancreatitis specimens from human patients, we employed well-defined animal models. We showed that treatment with cerulein in *Ptf1aCre^ERTM^* and *Ptf1aCre^ERTM^;LSL-KRAS^G12D^* mice showed downregulation of miR-15a and upregulation of two of its known targets: YAP1 and BCL-2 (Figure 1 and Figure 2). We demonstrated that *Ptf1aCre^ERTM^;LSL-KRAS^G12D^* mice treated with cerulein for four weeks have significantly increased fibrosis and that some cells positive for αSMA have proliferative status (Ki-67^+^) (Figure 3). Our in vitro studies determined that 5-FU-miR-15a suppresses the activation of pancreatic stellate cells (PSCs), as shown by reduced viability, proliferation, and migration (Figure 4, Figure 5, Figure 6, Figure 7, Figure 8, Figure 9 and Figure 10).

Specifically, we showed that 5-FU-miR-15a significantly inhibits the expression of *Yap1*, *Bcl2*, *Il6*, and *Mmp9* alone and during treatment with TGFβ1 in murine PSCs (Figure 9). The latter is particularly significant as it shows that 5-FU-miR-15a abrogates the role of TGFβ1 during PSCs activation. RT-PCR analysis only showed a slight increase in *Yap1* levels after TGFβ1 treatment at 24 h, and no increase at 48 h. In addition, we did not observe any substantial increase in the protein level of YAP1 upon TGFβ1. This is in agreement with previous data demonstrating that TGFβ1 has no impact on the status of YAP1. However, the crosstalk between YAP1/ TGFβ1 with YAP1 regulates the fibroinflammatory response of PSCs due to TGFβ1 stimulation [11]. Other studies showed that YAP-TEAD activation requires TGFβ1 signaling in cancer-associated fibroblasts and pancreatic stellate cell myofibroblasts [64,65]. Similarly, BCL-2 levels have been shown to not be affected or transiently affected by TGFβ1 signaling [66,67,68,69]. It is crucial that similar results regarding YAP1 and BCL2 reduction upon 5-FU-miR-15a treatment were obtained from murine and human PSCs (Figure 11). Regarding *Yap1* and *Bcl2*, our results are consistent with previous studies, showing the inhibitory effect of miR-15a and 5-FU-miR-15a on these factors’ expression [29,31]. Importantly, we showed that condition media obtained from PSCs treated with 5-FU-miR-15a or TGFβ1, and 5-FU-miR-15a significantly inhibit the invasion of pancreatic cancer cells (Figure 8). These results, and previously published data regarding pancreatic cancer cells, show that treatment with 5-FU-miR-15a may simultaneously inhibit the proliferation and migration of pancreatic stellate cells and pancreatic cancer cells [29]. Further studies employing three-dimensional co-culture studies of pancreatic stellate cells and pancreatic cancer cells should shed light on the efficacy of 5-FU-miR-15a in inhibiting the proliferation and migration of both types of cells.

Furthermore, recent studies showed that multiple miRs (miR-301, miR-200c, miR-149, miR-139, miR-34b) can affect the activity of PSCs during chronic pancreatitis, and modulation of their levels leads to PSCs apoptosis [70,71,72,73,74]. Moreover, studies demonstrated that targeting PSCs with antifibrotic or anti-inflammatory compounds in a form of siRNA or small molecules could prevent chronic pancreatitis progression [75,76,77,78,79,80,81,82]. YAP1 has been shown to induce the activity of PSCs and stimulate pancreatic fibrosis during chronic pancreatitis [83]. Spanehl and colleagues showed that siRNA against *Yap1* or verteporfin (YAP1 inhibitor) inhibits the proliferation and expression of fibrotic markers in activated rat PSCs [83]. HDAC inhibitors have been shown to increase the levels of miR-15 and miR-16, and lead to PSCs apoptosis and reduction in pancreatic fibrosis [84]. In vitro studies in activated rat PSCs showed that restoring the levels of miR-15b and miR-16 reduces *Bcl2* and leads to apoptosis [27]. Notably, our results showed that 5-FU-miR-15a suppresses inflammatory and fibrotic markers (*Mmp9* and *Il6*), thus providing a broader targeting pattern. In summary, our results demonstrate that ectopic delivery of 5-FU-miR-15a mimetics is a promising therapeutic approach for pancreatic fibrosis.

## 4. Materials and Methods

### 4.1. Animal Study

All studies and procedures involving animal subjects were approved by the Stony Brook University Institutional Animal Care and Use Committee (IACUC) and conducted strictly following the approved animal handling protocol. *Ptf1aCre^ERTM^* (Jackson Laboratory, Bar Harbor, Maine; Stock number 019378) and *LSL-Kras^G12D^* (Jackson Laboratory, Stock Number: 008179) mice have been described previously. All mice were maintained on a mixed background. *Ptf1aCre^ERTM^;LSL-Kras^G12D^* mice were generated by crossing *Ptf1aCre^ERTM^* and *LSL-Kras^G12D^* mice. The activity of Cre recombinase was induced by three intraperitoneal (IP) injections of tamoxifen (3 mg/injection, Millipore-Sigma, Burlington, MA, USA Cat. T5648-5G) dissolved in corn oil (Millipore-Sigma, Cat. C8267-2.5L) on alternating days. The chronic pancreatitis was induced by repeated hourly IP injections of cerulein (CER, Bachem Americas, Torrance, CA, USA, Cat. 4030451.0005) at a dose of 50 µg/kg body weight, dissolved in PBS (ThermoFisher Scientific, Pittsburgh, PA, USA, Cat. 21-031-CV), for 6 h three times a week for four weeks on alternating days. PBS was used as a control treatment.

### 4.2. Cell Lines

Murine pancreatic stellate cells were obtained from the laboratory of Dr. Means [85]. Pancreatic stellate cells (PSCs) were grown in a PSC medium consisting of RPMI 1640 (Fisher Scientific, Cat. 10-040-CV) containing 10% FBS (Peak Serum, Inc., Wellington CO, USA, Cat. PS-FB3), 5 μg/mL insulin, 5 μg/mL transferrin, 5 ng/mL sodium selenite (ThermoFisher Scientific, Pittsburgh, PA, USA, Cat. 41400045), 10^−6^ M hydrocortisone (Millipore-Sigma, Cat. H0888-5G), 100 U/mL penicillin (ThermoFisher Scientific, Cat. 15140163), 100 μg/mL streptomycin (ThermoFisher Scientific, Cat. 15140163), and 5 U/mL interferon-γ (Peprotech, Cranbury, NJ, USA, Cat. 315-05-100ug). Human pancreatic stellate cells were purchased from Creative Bioarray (Cat. CSC-7740W), CelPorgen (Cat. 36115-46), and ScienCell (Cat. 3830) and were grown under conditions specified by each supplier. UN-KPC-6141, a mouse pancreatic cancer cell line derived from KC (*Pdx1-Cre;LSL-Kras^G12D^*), was obtained from Dr. Surinder Batra [86]. UN-KPC-6141 were maintained in DMEM (Fisher Scientific, Cat. 10-013-CV), medium supplemented with 10% FBS (Peak Serum, Inc., Cat. PS-FB3), and 1% penicillin/streptomycin (ThermoFisher Scientific, Cat. 15140163).

### 4.3. Compounds

TGFβ1 was purchased from R&D Biosystems, Minneapolis, MN, USA (Cat. 7754-BH) and fluorouracil (5-FU) from Selleck Chemicals, Houston, TX, USA, (Cat. S1209). Control miR and miR-15a were obtained from ThermoFisher Scientific (Cat. AM17111 and AM17100, respectively), while 5-FU-miR-15a was generated by Horizon Discovery (CTM-628208 and CTM-628207). Lipofectamine 2000 Transfection Reagent was purchased from ThermoFisher Scientific (Cat. 11668027) and used according to the manufacturer’s protocol.

### 4.4. Cell Viability and Proliferation Assays

For cell viability experiments, PSCs were seeded at 10^3^ cell density in 100 μL of a medium in a 96-well white microplate and treated with variable concentrations of TGFβ1, 5-FU, CTRL miR, miR-15a, and 5-FU-miR-15a. Cell viability was assessed six days after the treatments using Cell-Titer Glo (Promega, Madison, WI, USA, Cat. G7571) assay according to the manufacturer’s protocol.

For cell proliferation experiments, PSCs were seeded at 2 × 10^5^ cell density in 2 mL of a medium in a 6-well microplate and treated with 5-FU, CTRL miR, miR-15a, and 5-FU-miR-15a at 50 nM concentration alone or in combination with TGFβ1 at 5 ng/mL. The cells were collected at 24, 48, 96, and 144 h after the treatment and counted using a Z2 Cell Counter (Beckman Coulter, Inc., Brea, CA, USA).

### 4.5. Cell Migration

For cell migration experiments, PSCs were seeded at 2 × 10^5^ cell density in 2 mL of a medium in a 6-well microplate and treated with DMSO, 5-FU, CTRL miR, miR-15a, and 5-FU-miR-15a at a 50 nM concentration alone and in combination with TGFβ1 at 5ng/mL for 24 h. Subsequently, a scratch was made using the tip of a 1000 μL pipette. Cell migration was evaluated at 12 and 22 h after the scratch using Nikon Eclipse Ti2 (Nikon Instruments, Inc., Melville, NY, USA) and analyzed using ImageJ Software [87].

### 4.6. Cell Invasion

PSCs were seeded at 2 × 10^5^ cell density in 2 mL of a medium in a 6-well microplate and treated with DMSO, 5-FU, CTRL miR, miR-15a, and 5-FU-miR-15a alone at a 50 nM concentration and in combination with TGFβ1 at 5 ng/mL for 24 h. Pancreatic cancer cells were seeded in CultreCoat 96well Low BME plate (R&D Systems, Cat. 3481-096-K) on top of the transwell, and conditioned media obtained from PSCs treatments were added to the bottom of the chambers. The plate was incubated for 22 h at 37 °C in a CO_2_ incubator. The measurements were done according to the manufacturer’s protocol.

### 4.7. Immunohistochemistry and Immunofluorescence

Pancreata dissected from mice were processed as previously described [88,89]. Tissue sections were first incubated with blocking buffer (5% BSA in TBS-Tween) for 30 min at 37 °C. Next, for immunohistochemistry staining slides were incubated with primary antibodies: anti-YAP1 (ThermoFisher Scientific, Cat. PA-46198) and anti-BCL-2 (Novus Biologicals, Centennial, CO, USA, Cat. NB10056098T) at 4 °C overnight in a humidified chamber with gentle shaking. Next, sections were washed and incubated with MACH 3 Rabbit Polymer Detection kit (Biocare Medical, Pacheco, CA, USA, Cat. M3R531H) for 30 min at 37 °C. Betazoid DAB (Biocare Medical, Cat. BDB2004L) was used to reveal IHC staining in tissues. For immunofluorescence staining, slides were incubated with primary antibodies: anti-αSMA (Abcam, Waltham, MA, USA, Cat. ab124964) and anti-Ki-67 (DAKO, Santa Clara, CA, USA Cat. M7249) were added and tissues were incubated at 4 °C overnight. The following day, secondary Alexa Fluor–labeled secondary antibodies (goat anti-rabbit 488, and goat anti-rat AF 647, ThermoFisher Scientific, Cat. A11008 and Jackson Immunoresearch, West Grove, PA, USA, Cat. 112-175-167, respectively) were added in blocking buffer for 30 min at 37 °C, counterstained with Hoechst 33258 (ThermoFisher Scientific, Cat. H3569), mounted with Fluoromount Aqueous Mounting Medium (Millipore-Sigma, Cat. F4680). Masson’s Trichrome staining was performed by the Research Histology Core, Stony Brook Cancer Center. Slides were analyzed under a Nikon Eclipse 90i microscope (Nikon Instruments, Inc.), and representative images were taken.

### 4.8. Western Blot

Total protein was extracted from cells with Laemmli buffer, and the analysis was performed as described previously [90]. Total protein was extracted from the frozen pancreas. Briefly, RIPA buffer (VWR, Radnor, PA, USA, Cat. 899000 with Halt (protease and phosphate inhibitor, ThermoFisher Scientific, Cat. 78440) was added to the frozen pancreas and disrupted using a rotor-stator homogenizer. The samples were centrifuged at 10,000× *g* for 10 min at 4 °C and the supernatant was collected and used for protein concentration assessment using BCA. Finally, samples were prepared with Laemmli buffer. Protein extracts were separated using 4–20% or 10% Criterion gels (Bio-Rad Laboratories, Hercules, CA, USA, Cat. 567-1093 and 567-1033, respectively), and transferred to the nitrocellulose membrane (Bio-Rad Laboratories, Cat. 162-0097). Primary antibodies rabbit anti-YAP1 (ThermoFisher Scientific, Cat. PA1-46198), rabbit anti-BCL-2 (Novus Biologicals, Cat. NB10056098T), mouse anti-ACTIN (Millipore-Sigma, Cat. A1978), goat anti-rabbit HRP-conjugated (Jackson ImmunoResearch, Cat. 111-035-144), and goat anti-mouse HRP-conjugated (Millipore-Sigma, Cat. AP200P) antibody were used. The images were developed using the Azure Biosystems (Dublin, CA, USA) imaging system (Azure 400).

### 4.9. Quantification of Western Blot

Densitometry analysis was performed using ImageJ software [87].

### 4.10. Reverse Transcription-Polymerase Chain Reaction (RT-PCR) Analysis

After appropriate treatments, total RNA from pancreatic stellate cells was used for quantitative PCR. According to the manufacturer’s instructions, RNA was extracted using PureLink RNA Mini Kit (ThermoFisher Scientific Cat. 12183018A). Primers against mouse *Yap1* (Mm01143263_m1), *Bcl2* (Mm00477631_m1), *Il6* (Mm00446190_m1), *Mmp9* (Mm00442991_m1), *miR-15a*, and *Hprt1* (Mm03024075_m1), and against human *YAP1* (Hs00902712_g1), *BCL2* (Hs010489321_g1), *IL6* (Hs00174131_m1), *MMP9* (Hs00957562_m1), and *HPRT1* (Hs02800695_m1) were purchased from ThermoFisher Scientific. The cDNA was prepared using SuperScript^®^ VILO™ cDNA Synthesis Kit (ThermoFisher Scientific, Cat. 11754-050). Analysis was performed with Applied Biosystems TaqMan™ Gene Expression Master Mix (ThermoFisher Scientific, Cat. 43-690-16) using QuantStudio3 (ThermoFisher Scientific) as per standard protocols. Observed CT values were then used to calculate fold change using the 2-ΔΔCt relative quantification method. Mouse *Hprt1* and human *HPRT1* were used as the housekeeping control genes.

### 4.11. Statistical Analysis

The analysis of in vitro experiments were performed using One-Way ANOVA or *t*-test with a value of *p* < 0.05 considered significant. This analysis was performed using GraphPad Prism version 9.4.1 for macOS (GraphPad Software, www.graphpad.com).

## 5. Conclusions

In summary, we showed that 5FU-modified miR-15a inhibits the expression of YAP1 and BCL-2 in murine and human PSCs in the in vitro setting. Importantly, we demonstrated that 5FU-modified miR-15a suppresses PSCs proliferation and migration, and inhibits the invasion of pancreatic cancer cells.

## 6. Patents

J.J have filed a patent for 5-FU-modified miRNA mimetics.

## Figures and Tables

**Figure 1 ijms-24-03954-f001:**
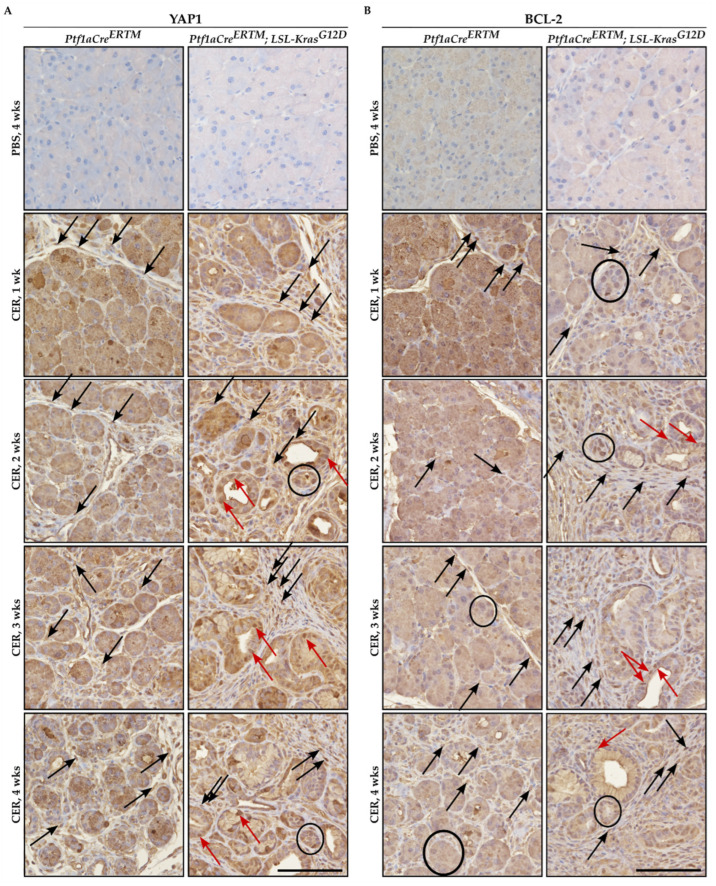
Immunohistochemical staining of YAP1 and BCL-2 during the development of chronic pancreatitis. Pancreatic specimens were obtained from *Ptf1aCre^ERTM^* and *Ptf1aCre^ERTM^;LSL-Kras^G12D^* mice were treated with tamoxifen for one week and subsequently with PBS (vehicle) and CER (cerulein) for four weeks and samples were collected 1, 2, 3, and 4 weeks after the injury. (**A**) Immunohistochemical stain of YAP1, and (**B**) immunohistochemical stain of BCL-2. The scale bar represents 100 µm. Black arrows point to fibrotic cells, black circles mark injured pancreatic acinar cells, and red arrows mark early pancreatic neoplastic cells positive for YAP1 or BCL-2 in Figure 1A and Figure 1B, respectively.

**Figure 2 ijms-24-03954-f002:**
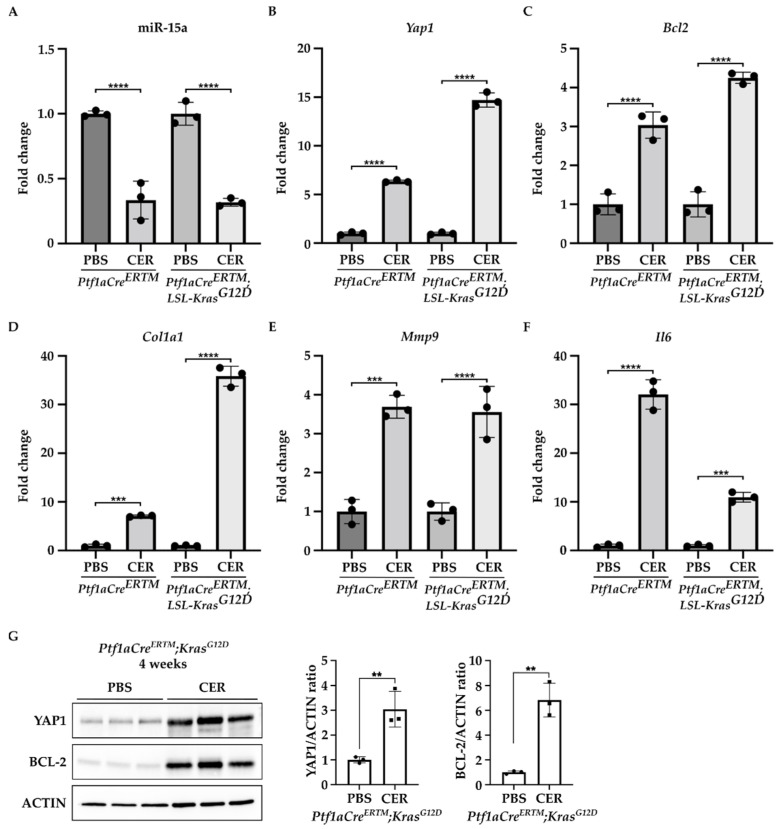
YAP1 and BCL-2 levels are upregulated, and miR-15a levels are downregulated during the development of chronic pancreatitis. Pancreatic specimens were obtained from *Ptf1aCre^ERTM^* and *Ptf1aCre^ERTM^;LSL-Kras^G12D^* mice treated with tamoxifen for one week and subsequently with PBS (vehicle) or CER (cerulein) for four weeks. (**A**) Represents RT-PCR levels of miR-15. (**B**) Represents RT-PCR levels of *Yap1*. (**C**) Represents RT-PCR levels of *Bcl2*. (**D**) Represents RT-PCR levels of *Col1a1*. (**E**) Represent RT-PCR levels of *Mmp9*. (**F**) Represent RT-PCR levels of *Il6*. Data are represented as the mean ± SD; N = 3. *** *p* < 0.001 and **** *p* < 0.0001 by One-Way ANOVA. (**G**) Western blot analysis and quantification of YAP1 and BCL-2 levels in *Ptf1aCre^ERTM^;LSL-Kras^G12D^* mice treated with tamoxifen for one week and subsequently with PBS (vehicle) or CER (cerulein) for four weeks. ACTIN was used as a loading control. Data are represented as the mean ± SD; N = 3. ** *p* < 0.01 using *t*-test.

**Figure 3 ijms-24-03954-f003:**
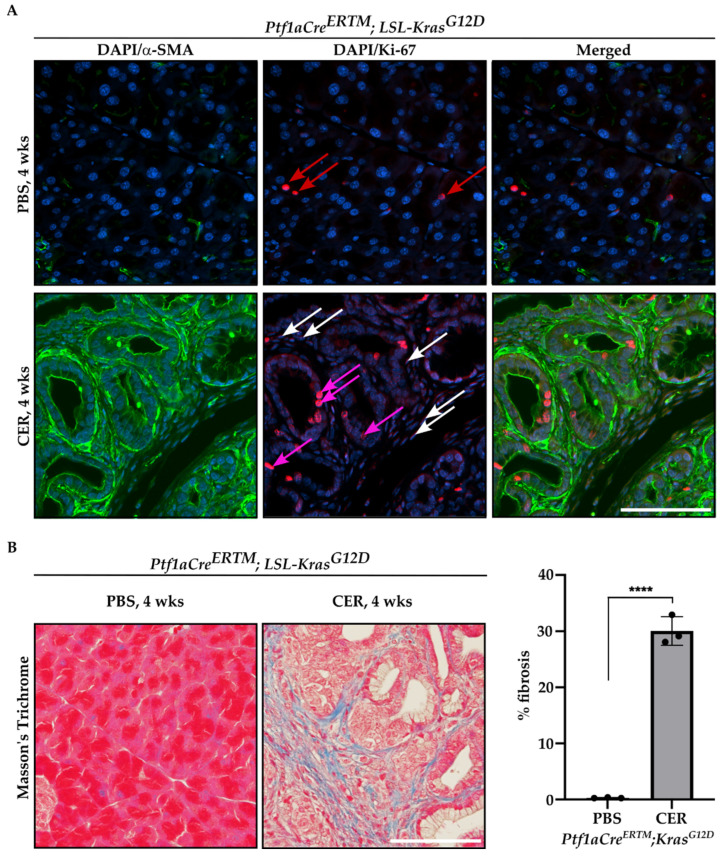
Proliferation and fibrosis are increased during the development of chronic pancreatitis. Pancreatic specimens were obtained from *Ptf1aCre^ERTM^;LSL-Kras^G12D^* mice treated with tamoxifen for one week and subsequently with PBS (vehicle) or CER (cerulein) for four weeks. (**A**) Immunofluorescence staining of DAPI/αSMA, DAPI/Ki-67, and DAPI/αSMA/Ki-67, red arrows mark Ki-67^+^ cells in pancreas obtained from *Ptf1aCre^ERTM^;LSL-Kras^G12D^* mice treated with PBS, white arrows mark stromal cells positive for Ki-67^+^ and pink arrows mark early pancreatic neoplastic cells positive for Ki-67^+^ in pancreas obtained from *Ptf1aCre^ERTM^;LSL-Kras^G12D^* mice treated with CER, and (**B**) representative Masson’s Trichrome staining with quantification. The scale bar represents 100 µm. Data are represented as the mean ± SD; N = 3. **** *p* < 0.0001 using *t*-test.

**Figure 4 ijms-24-03954-f004:**
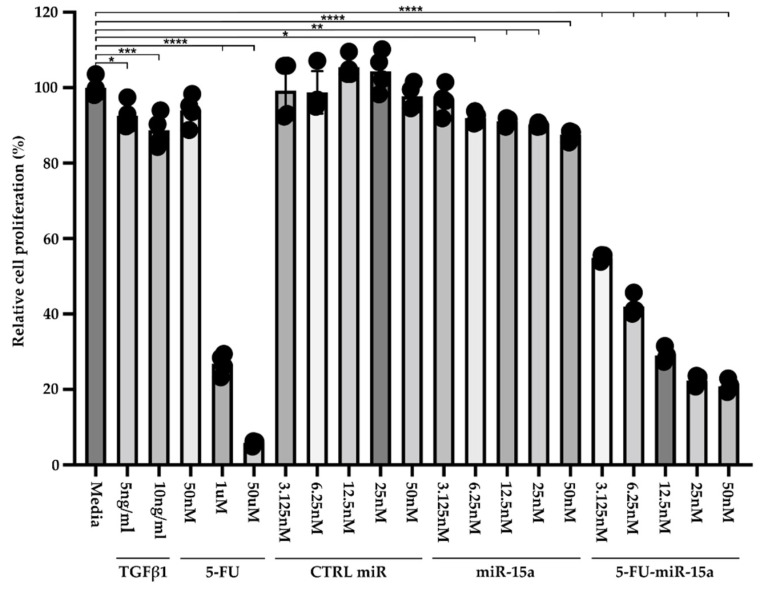
Dose-response treatment of murine pancreatic stellate cells. Pancreatic stellate cells were treated with TGFβ1, 5-FU, CTRL miR, miR-15a, and 5-FU-miR-15a for six days, and cell viability was assessed using Cell-Titer Glo and compared to the control (medium). Data are represented as the mean ± SD; N = 4. * *p* < 0.05, ** *p* < 0.01, *** *p* < 0.001, and **** *p* < 0.0001 by One-Way ANOVA.

**Figure 5 ijms-24-03954-f005:**
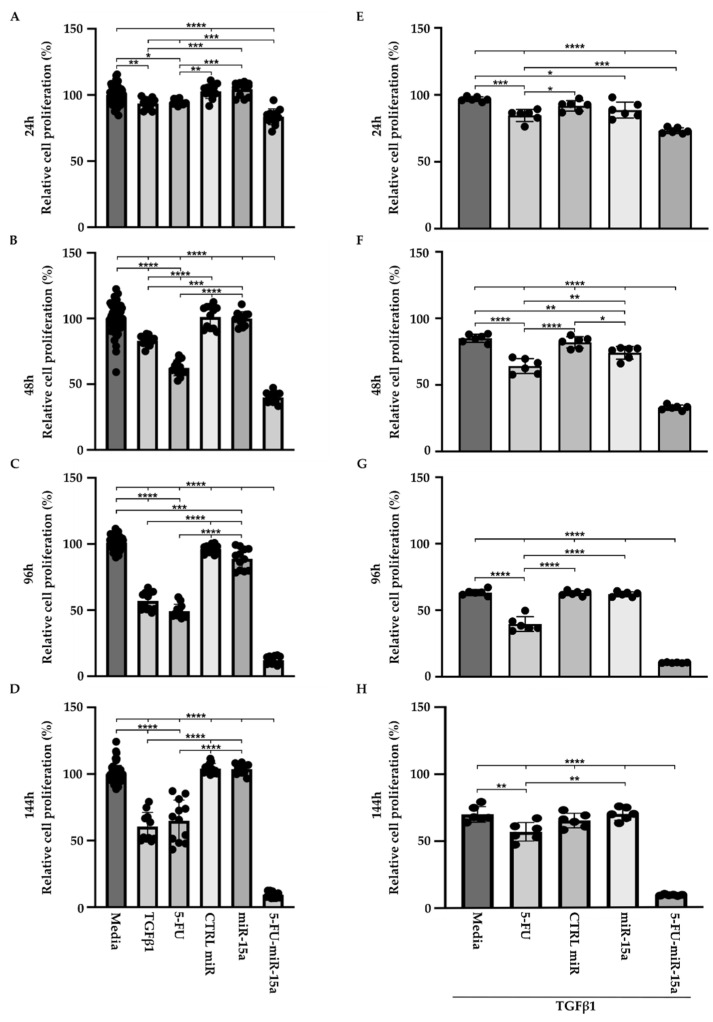
5-FU-miR-15a inhibits the growth of murine pancreatic stellate cells as a single agent and in the context of TGFβ1 treatment. Pancreatic stellate cells were treated with TGFβ1, and/or 5-FU, CTRL miR, miR-15a, and 5-FU-miR-15a. Cell viability was assessed at (**A**) 24 h, (**B**) 48 h, (**C**) 96 h, (**D**) 144 h after single agent treatment, and at (**E**) 24 h, (**F**) 48 h, (**G**) 96 h, (**H**) 144 h after co-treatment with TGFβ1. Data are represented as the mean ± SD; N = 12 for a single treatment and N = 6 for a double treatment. * *p* < 0.05, ** *p* < 0.01, *** *p* < 0.001, and **** *p* < 0.0001 by One-Way ANOVA.

**Figure 6 ijms-24-03954-f006:**
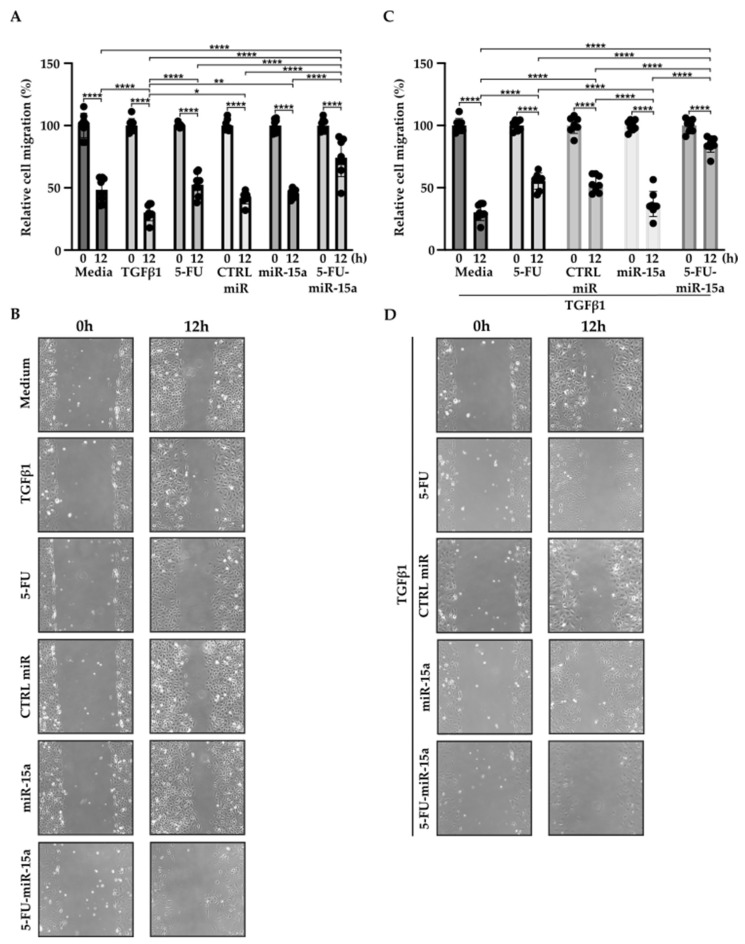
5-FU-miR-15a inhibits the migration of pancreatic stellate cells as a single agent and in the context of TGFβ1 treatment. Pancreatic stellate cells were treated with TGFβ1, and/or 5-FU, CTRL miR, miR-15a, and 5-FU-miR-15a. Migration was assessed 12 h post-scratch (**A**)–quantitative analysis and (**B**) representative images after single agent treatment, and (**C**)–quantitative analysis and (**D**) representative images after co-treatment with TGFβ1. Data are represented as the mean ± SD, N = 8. * *p* < 0.05, ** *p* < 0.01, and **** *p* < 0.0001 by One-Way ANOVA.

**Figure 7 ijms-24-03954-f007:**
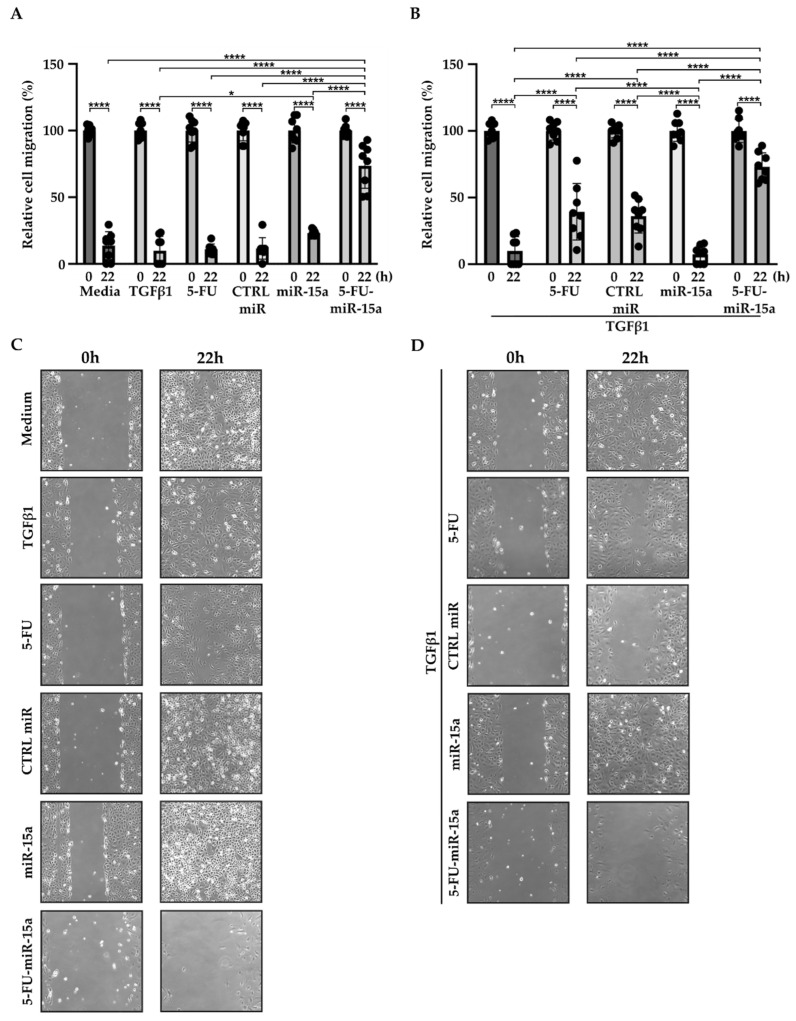
5-FU-miR-15a inhibits the migration of pancreatic stellate cells as a single agent and in the context of TGFβ1 treatment. Pancreatic stellate cells were treated with TGFβ1, and/or 5-FU, CTRL miR, miR-15a, and 5-FU-miR-15a. Migration was assessed 22 h post-scratch (**A**)–quantitative analysis and (**B**) representative images after single agent treatment, and (**C**)–quantitative analysis and (**D**) representative images after co-treatment with TGFβ1. Data are represented as the mean ± SD, N = 8. * *p* < 0.05 and **** *p* < 0.0001 by One-Way ANOVA.

**Figure 8 ijms-24-03954-f008:**
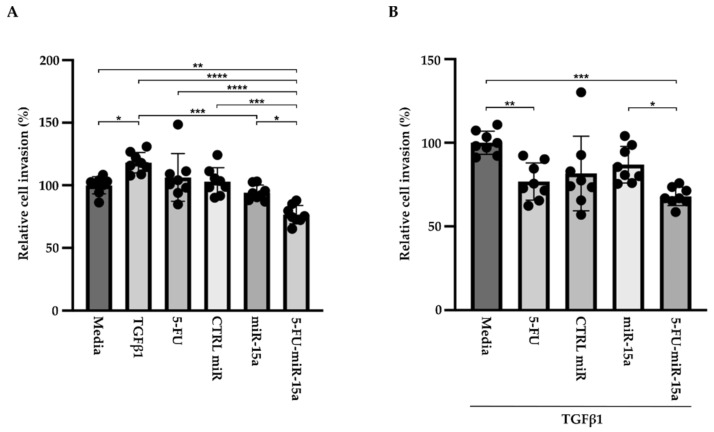
5-FU-miR-15a medium obtained from pancreatic stellate cells reduced invasion of pancreatic cancer cells (KPC). Pancreatic cancer cells were exposed to conditioned media obtained from pancreatic stellate cells treated with TGFβ1, and/or 5-FU, CTRL miR, miR-15a, and 5-FU-miR-15a. The invasion was assessed 22 h post-treatment with (**A**) media obtained from pancreatic stellate cells after single agent treatment and (**B**) media obtained from pancreatic stellate cells co-treated with TGFβ1. Data are represented as the mean ± SD, N = 8. * *p* < 0.05, ** *p* < 0.01, *** *p* < 0.001, and **** *p* < 0.0001 by one-way ANOVA.

**Figure 9 ijms-24-03954-f009:**
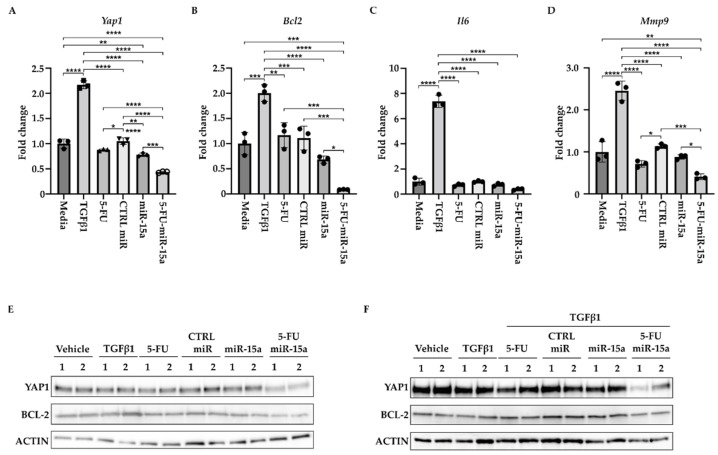
5-FU-miR-15a reduces the levels of expression of survival and inflammatory markers. PSCs were treated for 24 h under indicated conditions. (**A**) Represents RT-PCR levels of *Yap1*. (**B**) Represents RT-PCR levels of *Bcl2*. (**C**) Represents RT-PCR levels of *Mmp9*. (**D**) Represents RT-PCR levels of *Il6*. Data are represented as the mean  ±  SD Data are represented as the mean ± SD, N = 3. * *p* < 0.05, ** *p* < 0.01, *** *p* < 0.001, and **** *p* < 0.0001 by One-Way ANOVA. (**E**) Western blot analysis of PSCs after a single agent treatment. (**F**) Western blot analysis of PSCs after co-treatment with TGFβ1. Data are shown in duplicate.

**Figure 10 ijms-24-03954-f010:**
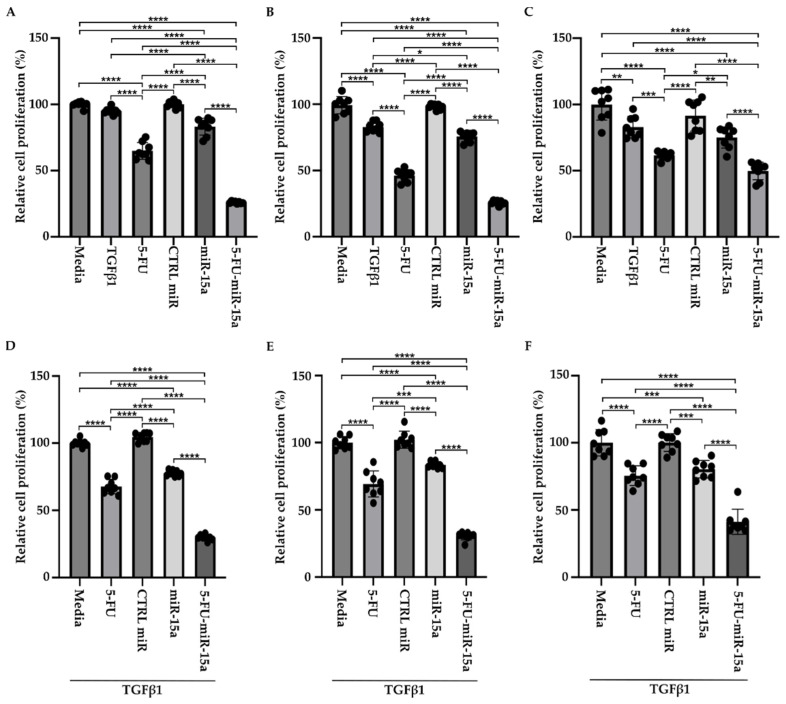
5-FU-miR-15a inhibits the growth of human pancreatic stellate cells as a single agent and in the context of TGFβ1 treatment. Human pancreatic stellate cells were treated with TGFβ1 and/or 5-FU, CTRL miR, miR-15a, and 5-FU-miR-15a, and cell viability was assessed at 144 h post-treatment. (**A**,**D**)—hPSCs obtained from Creative Bioarray, (**B**,**E**)—hPSCs obtained from ScienCell, and (**C**,**F**)—hPSCs obtained from CelProgen. Data are represented as the mean ± SD; N = 8 for single (**A**–**C**) and combinatorial (**D**–**F**) treatment. * *p* < 0.05, ** *p* < 0.01, *** *p* < 0.001, and **** *p* < 0.0001 by One-Way ANOVA.

**Figure 11 ijms-24-03954-f011:**
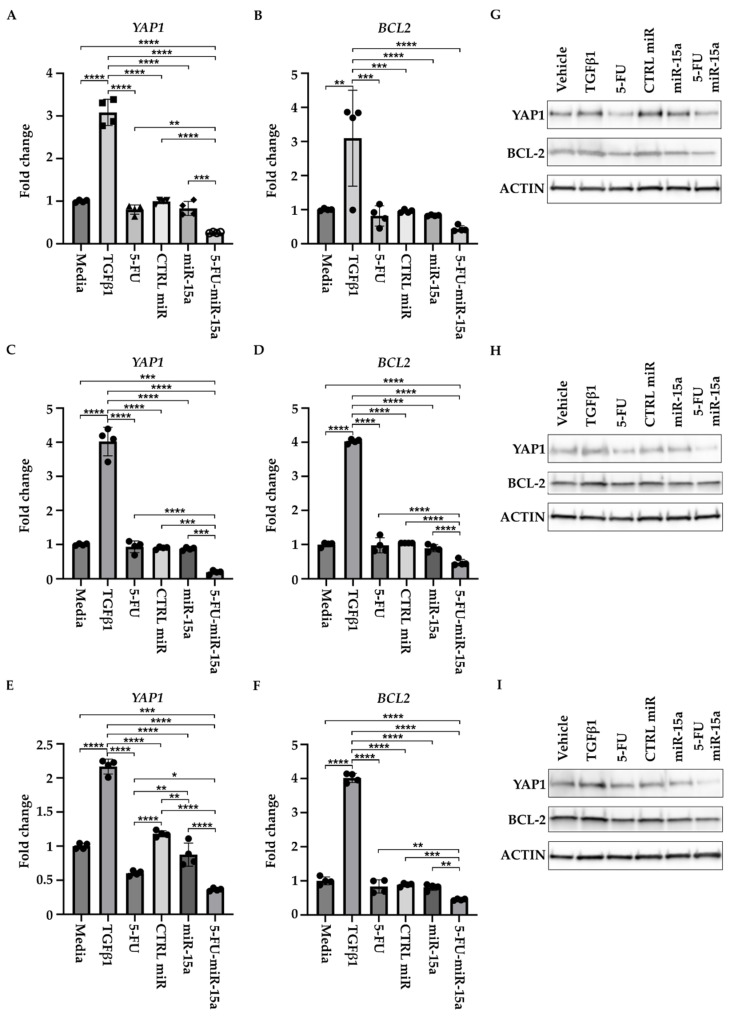
5-FU-miR-15a inhibits YAP1 and BCL-2 levels of human pancreatic stellate cells in vitro. Human pancreatic stellate cells were treated with TGFβ1 or 5-FU, CTRL miR, miR-15a, and 5-FU-miR-15a, and samples for RNA and protein analysis were collected 24 h post-treatment. RNA analysis of YAP1 and BCL2 levels. (**A**,**B**) hPSCs obtained from Creative Bioarray; (**C**,**D**) hPSCs obtained from CelProgen; (**E**,**F**) hPSCs obtained from ScienCell. Data are represented as the mean ± SD; N = 12 for a single treatment and N = 6 for a double treatment. * *p* < 0.05, ** *p* < 0.01, *** *p* < 0.001, and **** *p* < 0.0001 by one-way ANOVA. Western analysis of YAP1 and BCL2 levels (**G**) hPSCs obtained from Creative Bioarray; (**H**) hPSCs obtained from CelProgen; (**I**) hPSCs obtained from ScienCell.

## Data Availability

Not applicable.

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
