# Peer review of "5-FU-miR-15a Inhibits Activation of Pancreatic Stellate Cells by Reducing YAP1 and BCL-2 Levels In Vitro"

_ijms, 2023, doi:10.3390/ijms24043954_

Round 1

Reviewer 1 Report

5-FU-miR-15a inhibits activation of pancreatic stellate cells by 2reducing YAP1 and BCL-2 levels in vitro

The authors showed that 5FU-modified miR15a suppresses PSCs proliferation and migration by which inhibiting the invasion of pancreatic cancer cells. In vitro, 5FU-modified miR15a inhibits the expression of YAP1 and BCL-2 in murine and human PSCs. Thus, this manuscript is very important. However, there are some concerns at this point.

Major comments

1.      It should be appreciated if I could see western blots of YAP1 and BCL-2 obtained from the pancreas of Ptf1aCreERTM;LSL-KrasG12D mice.

2.      It should be appreciated if I could see MIB-1 (Ki-67 staining) for stellae cells obtained from the pancreatic specimens of Ptf1aCreERTM;LSL-KrasG12D mice.

Minor comment

1.      Quantification of pancreatic fibrosis obtained from the pancreas of Ptf1aCreERTM;LSL-KrasG12D mice would be shown.

Author Response

February 9th, 2023

Editor and Reviewers

Dear Editor and Reviewers,

We want to thank the Reviewers for their invaluable comments. We also acknowledge the concerns raised by the Reviewers and now provide a revised copy of the manuscript, considering all of the Reviewers' concerns. This revised manuscript addressed all the comments and suggestions the Editor and Reviewers provided. Please see our responses to your comments below.

Reviewer 1.

Comment 1. It should be appreciated if I could see western blots of YAP1 and BCL-2 obtained from the pancreas of Ptf1aCreERTM;LSL-KrasG12D mice.

Response 1. We provided western blot analysis of protein extracts obtained from Ptf1aCreERTM;LSL-KrasG12D after four weeks of treatment with PBS or cerulein (CER). This figure and the quantification of the results are included as part of Figure 2.

Comment 2. It should be appreciated if I could see MIB-1 (Ki-67 staining) for stellate cells obtained from the pancreatic specimens of Ptf1aCreERTM;LSL-KrasG12D mice.

Response 2. We included immunofluorescence co-staining of aSMA/Ki-67/DAPI performed on pancreatic tissues extracts obtained from Ptf1aCreERTM;LSL-KrasG12D after four weeks of treatment with PBS or cerulein (CER). We included aSMA/DAPI and Ki-67/DAPI and merged images of all these three stainings. These results are presented in Figure 3A.

Comment 3. Quantification of pancreatic fibrosis obtained from the pancreas of Ptf1aCreERTM;LSL-KrasG12D mice would be shown.

Response 3. We performed Masson’s Trichrome staining on on pancreatic tissue obtained from Ptf1aCreERTM;LSL-KrasG12D after four weeks of treatment with PBS or cerulein (CER). We included representative images from each treatment and quantification. These results are included in Figure 3B.

Reviewer 2.

Comments 1 and 2. The foundation of the paper is Fig. 1. The concrete evidence that YAP1 and BCL-2 are up-regulated in chronic pancreatitis should emerge from the images in Fig. 1A & 1B. Unfortunately, these images - at a far too low magnification and resolution - are not at all convincing and are not properly labelled. As such, it is difficult to understand what precisely we are supposed to see. For example, the authors claim (lines 95-97) that the increase of both YAP1 and BCL-2 was observed in injured acinar cells and that this can be seen in Figs. 1A and B. If so, they should show this at a much higher level of magnification and, in the normal way of presenting such data, signpost with arrows exactly what is significant. Importantly, one should also be able to see what happens in the stellate cells, since these are supposed to be the focus of this paper! However, this point is completely unclear.

Furthermore, given that this part of the study is critical for the underpinning of the whole story, it would also be much more convincing if the changes in YAP1 and BCL-2 levels were assessed not just at one single time point, as in the present ms, but monitored over time, as chronic pancreatitis gradually develops.

Responses 1 and 2. We provided images obtained from immunohostochemical stain of pancreata from Ptf1aCreERTM and Ptf1aCreERTM;LSL-KrasG12D after one, two, three, and four weeks of cerulein treatment and four weeks of PBS treatment. We included higher magnifications of all images and marked using arrows: activated pancreatic stellate cells (fibroblasts), early pancreatic neoplasia, and injured pancreatic acinar cells that undergo acinar-to-ductal metaplasia. These images are included in Figure 1.

Comment 3. The context in which the data are introduced and later discussed needs to be improved. Lines 41-45, for example, rely on references that are not up-to-date and/or not comprehensive. The authors should consult a more recent and comprehensive review (Physiol Rev 101: 1691-1744, 2021).

Response 3. We updated our discussion and included a more recent review suggested by the Reviewer.

Furthermore, we modified the text to improve the English language and style.

We hope the incorporated changes will satisfy the Reviewers and Editor and render the revised manuscript suitable for publication. Thank you for being so considerate.

We confirm that neither the manuscript nor any parts of its content are currently under consideration or published in another journal.

All authors have approved the manuscript and agree with its submission to the International Journal of Molecular Sciences.

Please feel free to contact me if I can be of further assistance,

Sincerely Yours,

Agnieszka B. Bialkowska, PhD

Associate Professor

Renaissance School of Medicine at Stony Brook University

Department of Medicine

GI Translational Research Lab

HSC-T17 Room 090

Stony Brook, NY 11794-8176

Phone: (631) 638 2161

Email: Agnieszka.Bialkowska@stonybrookmedicine.edu

Reviewer 2 Report

This paper deals with an interesting and important issue. If the data description can be significantly improved to make the results convincing and if the results can then be placed in the proper contemporary context, this could become a valuable contribution. There are two critical points that need to be addressed:

The foundation of the paper is Fig. 1. The concrete evidence that YAP1 and BCL-2 are up-regulated in chronic pancreatitis should emerge from the images in Fig. 1A & 1B. Unfortunately, these images - at a far too low magnification and resolution - are not at all convincing and are not properly labelled. As such, it is difficult to understand what precisely we are supposed to see. For example, the authors claim (lines 95-97) that the increase of both YAP1 and BCL-2 was observed in injured acinar cells and that this can be seen in Figs. 1A and B. If so, they should show this at a much higher level of magnification and, in the normal way of presenting such data, signpost with arrows exactly what is significant. Importantly, one should also be able to see what happens in the stellate cells, since these are supposed to be the focus of this paper! However, this point is completely unclear. Furthermore, given that this part of the study is critical for the underpinning of the whole story, it would also be much more convincing if the changes in YAP1 and BCL-2 levels were assessed not just at one single time point, as in the present ms, but monitored over time, as chronic pancreatitis gradually develops.

The context in which the data are introduced and later discussed needs to be improved. Lines 41-45, for example, rely on references that are not up-to-date and/or not comprehensive. The authors should consult a more recent and comprehensive review (Physiol Rev 101: 1691-1744, 2021).

Author Response

(The authors gave the same response as above.)

Round 2

Reviewer 1 Report

​The authors have precisely answered, so I accept this manuscript.

Reviewer 2 Report

The revision has been successful and the issues I had raised in my original report have been resolved satisfactorily.